# A Pilot Study Using Entropy for Optimizing Self-Pacing during a Marathon

**DOI:** 10.3390/e25081119

**Published:** 2023-07-26

**Authors:** Florent Palacin, Luc Poinsard, Jean Renaud Pycke, Véronique Billat

**Affiliations:** 1Laboratory of Neurophysiology and Movement Biomechanics, Université Libre de Bruxelles Neuroscience Institut, 1070 Bruxelles, Belgium; luc.poinsard@gmail.com; 2Billatraining SAS, 91840 Soisy-sur-École, France; 3UMR8071-CNRS-Laboratoire de Mathématiques et Modélisation d’Evry, Université Paris-Saclay, Univ Evry, 91000 Evry-Courcouronnes, France; jeanrenaud.pycke@univ-evry.fr; 4EA 4526-Laboratoire IBISC Paris-Saclay, Univ Evry, 91000 Evry-Courcouronnes, France; veroniquelouisebillat@gmail.com

**Keywords:** marathon running, hitting the wall, Shannon entropy, stride length, performance

## Abstract

A new group of marathon participants with minimal prior experience encounters the phenomenon known as “hitting the wall,” characterized by a notable decline in velocity accompanied by the heightened perception of fatigue (rate of perceived exertion, RPE). Previous research has suggested that successfully completing a marathon requires self-pacing according to RPE rather than attempting to maintain a constant speed or heart rate. However, it remains unclear how runners can self-pace their races based on the signals received from their physiological and mechanical running parameters. This study aims to investigate the relationship between the amount of information conveyed in a message or signal, RPE, and performance. It is hypothesized that a reduction in physiological or mechanical information (quantified by Shannon Entropy) affects performance. The entropy of heart rate, speed, and stride length was calculated for each kilometer of the race. The results showed that stride length had the highest entropy among the variables, and a reduction in its entropy to less than 50% of its maximum value (H = 3.3) was strongly associated with the distance (between 22 and 40) at which participants reported “hard exertion” (as indicated by an RPE of 15) and their performance (*p* < 0.001). These findings suggest that integrating stride length’s Entropy feedback into new cardioGPS watches could improve marathon runners’ performance.

## 1. Introduction

Paris 2024 is set to create history by offering the public a unique opportunity to participate in the Olympic Marathon, following the exact same route as the elite athletes. The Mass Participation Marathon, also known as “Marathon Pour Tous”, has no entry fee and provides 2024 slots for participants. This democratization of the marathon is leading to the emergence of new marathon runners, many of whom experience physical duress commonly referred to as “hitting the wall” (HTW) [1,2]. HTW is primarily influenced by running strategy, specifically by maintaining an excessively fast pace during the initial half marathon [3,4,5]. The most prevalent physiological indicators associated with HTW are overall fatigue and fatigue localized in the legs [1].

Research has demonstrated that muscular power output is regulated in an anticipatory manner to prevent uncontrolled disruptions in physiological homeostasis [6]. Pacing has a significant impact on energy production from both aerobic and anaerobic energy systems. The goal of the pacing strategy is to optimize these energy systems accordingly. Although the effects of various physiological regulators overlap, the conscious brain integrates their net input using the rating of perceived exertion (RPE) [7,8,9,10]. Changes in homeostatic status, reflected by momentary RPE, allow for alteration of pacing strategy (power output) in both an anticipatory and responsive manner based on pre-exercise expectations and peripheral feedback from different physiological sensors [11,12]. Recent studies have examined the continuous physiological response and RPE during marathons, revealing a similar decrease in the ratio between RPE and speed, heart rate, and $\dot{V}$O_2_ for all recreational runners [4].

Our current understanding of how runners adapt their marathon pace to account for various cardiorespiratory and biomechanical factors remains incomplete. One hypothesis proposes a strong connection between the rating of perceived exertion (RPE) and a physiological and mechanical message that is crucial for maintaining accurate adjustments in running speed. More specifically, it is essential to achieve a balance between stride amplitude and stride frequency, in the same way that a cyclist needs to dose his equipment. Furthermore, the signal must possess a sufficient level of uncertainty to effectively convey information.

The concept of information, as originally defined by Shannon [13], represents a physical quantity associated with systems capable of existing in multiple states. All life-related functions can be described in physical terms as information processing. The sensory organs of higher organisms perceive information from the environment and transmit it to the nervous system for further processing. The neural network then facilitates the exchange of information between the different parts of the organism and the central nervous system. Living organisms rely on a constant supply of information, which they continually process to maintain their organized structure. Without this flow of information, which is necessary for their various functions, the structured organization of living organisms would gradually disintegrate as their constituent materials undergo irreversible changes towards a state of randomness and disorganization. Consequently, living organisms can be viewed as reservoirs of physical information that is constantly maintained and transformed.

Entropy plays a pivotal role in this context. It serves as a measure of the level of uncertainty or randomness within a given dataset. In information theory, entropy is measured in bits and calculated based on the probabilities of the various possible outcomes. If a message contains a set of possible outcomes with equal probabilities, the entropy of the message will be high due to a significant degree of uncertainty or randomness. Conversely, if the message contains only one possible outcome, the entropy will be low because there is no uncertainty or randomness. We propose a hypothesis that suggests marathon runners experience the phenomenon known as “hitting the wall” when their physiological and mechanical parameters fall below a threshold associated with their maximum entropy, as determined through a maximal test. This threshold could represent an entropy reserve that enables runners to remain aware and adjust their speed accordingly.

The objective of this study is twofold: (1) to examine whether entropy values derived from heart rate (HR), stride length (SL), or speed (which are currently accessible through cardio-GPS watches) can serve as indicators of an increase in the rating of perceived exertion (RPE), and (2) to determine the level of entropy decrease in HR, SL, or speed (referred to as the “entropy threshold decrease”) at which marathon runners typically experience hitting the wall.

## 2. Materials and Methods

### 2.1. Subject

Six non-elite male marathoners with the following characteristics (mean ± standard deviation (SD)): age: 37 ± 8.8 years, weight: 73.4 ± 6.1 kg, and height: 179.3 ± 3.6 cm participated in this study (Table 1). For organizational reasons, we chose to have only one gender in the present study in order to avoid introducing an additional factor that could influence the statistical analysis. All the subjects volunteered to participate in the study and were asked not to modify their usual training.

They were selected for their homogeneous physiological and endurance characteristics [14,15,16], and all runners had already run at least two marathons. All subjects reported training three to four times per week (50–80 km/week) for more than 5 years. Once a week, they performed high-intensity interval training of 6 × 1000 m at 90–100% of their maximal heart rate and tempo training (15–25 km) at 100–90% of their average marathon speed. The study’s objectives and procedures were approved by an institutional review board (CPP Sud-Est V, Grenoble, France; reference: 2018-A01496-49). All participants received information about the study and gave their written consent to participate.

### 2.2. Experimental Design: RABIT^®^ Test and the Marathon Race

All participants performed the RABIT^®^ (Running Advisor Billat Training) test, to determine individual $\dot{V}$O_2_max and HRmax values. Three days later, all participants ran a marathon in an official race (Annecy Lake Marathon, France). The race started at 9 a.m. The temperature was between 6 and 8 °C (between 9 a.m. and 1 p.m.), with no precipitation and an average humidity of 55%. Blood lactate was measured on the finger (Lactate PRO2 LT-1730; ArKray, Japan) just after the warm-up (15 min at an easy pace) and three minutes after crossing the finish line.

The RABIT^®^ test was performed outdoors using a portable gas exchange system to determine $\dot{V}$O_2_max and HRmax. The RABIT^®^ test has been validated as a valid field test of maximal and functional aerobic capacity [17,18]. This RABIT^®^ test consisted of 3 incremental exercise stages, adjusted to a prescribed RPE (rating of perceived exertion) [7] equivalent to “light” (RPE 11) for 10 min, “somewhat hard” (RPE 14) for 5 min, and “very hard” (RPE 17) for 3 min. Exercise intensity is assessed subjectively by the person exercising. The corresponding written descriptions range from “very light” to “very, very hard”. The scale correlates well with cardiorespiratory and metabolic variables such as minute ventilation, heart rate, and blood lactate levels [7]. Each step was followed by a 1 min rest period. Participants were instructed to modify their running speed on a moment-to-moment basis in line with the prescribed RPE (rather than the endpoint of the task), so that their RPE (not their speed) remained constant for each stage. The test was conducted outdoors on a hard dirt path. The RPE scale could be consulted by the participant at regular intervals (i.e., every 100 m) because he was followed by the investigator on a bike.

### 2.3. Experimental Measurements

RABIT^®^ Test

Respiratory gases (oxygen uptake, $\dot{V}$O_2_) were continuously measured using a telemetric, portable, breath-by-breath sampling system (K5, Cosmed, Rome, Italy). A GPS running watch (Garmin [19], Olathe, KS, USA) paired with the K5 [20] was used to measure heart rate (HR), stride length (SL), and the speed (V) response (using 5 s data averages) throughout each trial. We used the same cardiac belt for the Garmin and K5 because it was compatible for both.


Marathon


The same watch used for the RABIT^®^ test was used for the marathon to measure V, SL, and HR. Given that performance in a marathon has recently been shown to be dependent on pacing oscillations [21], we encouraged runners to self-pace their run without focusing on the cardio-GPS whose dial was hidden. During the marathon, refreshment points (offering water, dry and fresh fruit, and sugar) were located every 5 km and at the finish line. Sponge stations were located every 5 km from km 7.5 Runners were allowed to remove their masks at the refreshment points to eat and drink. All runners were given a drink at each refreshment point (with flat water and fruits). The RPE was recorded by runners using a small microphone attached to the jersey. Runners recorded an RPE at least every km or more frequently if they felt the need. We used the Borg 6–20 scale [7] to assess fatigue during the marathon in correlation with physiological stress indicators. Runners were familiarized with the scale in the two weeks prior to the race.

### 2.4. Data Analysis

We analyzed the variables V, HR, RPE, and SL for every kilometer of the marathon. We consider that there is a steady state change when the entropy value differs from the mean by ±2 standard deviation. We call this phenomenon the entropy falling.

### 2.5. Statistical Analysis


Shannon’s Entropy


The Shannon entropy is an indicator of the average information embedded in a message. In fact, in this research, a message is considered a time series. A higher Shannon entropy indicates that the information contained in a message is more important [22]. The entropy H(x) of a discrete random variable X is a measure of its average uncertainty. Entropy is calculated by the equation [23]:(1)$$H\left(X\right)=-\underset{x_{i}}{\sum }p(x_{i})\log_{2}\left(x_{i}\right)$$where in this research, $p\left(x_{i}\right)$ indicates the probability of a state (V, HR, SL) with the value of $x_{i}$, where i can be change along with the signal. In our case, we have 9 states defined in our RABIT^®^ test. We calculated the quartiles (Q) for the 3 running sensations, namely easy, medium, and hard. Each sensation is divided into 4 equal parts that each represent 25% of the area. In this study, we only used Q1, Q2, and Q3 which delimit 2 zones for each sensation. States 1 and 2 represent the two zones of easy pace (RPE 11), states 4 and 5 for medium pace (RPE 14), and states 7 and 8 for hard pace (RPE 17). States 0, 3, 6 and 9 represent the values between our three incremental exercise steps, as shown in the graph below (Figure 1).

Entropy was calculated for each kilometer of the marathon. The maximum entropy (*H*_max_) is calculated according to the following equation:(2)$$H_{\max}=\log_{2}\left(N\right)$$
where *N* indicates the number of possible states in the signal (*N* = 9).

Therefore, to illustrate how entropy is calculated for speed, the values are quantified in nine states according to the RPE values obtained in the RABIT^®^ test (Figure 1, Table 2).


Global Tendency of Pace and Its Asymmetry


The coefficients of variation are calculated using all V, HR, and SL values, i.e., the values calculated every 5 s.

The trend in speed time series (i.e., Kendall’s *τ* non-parametric rank correlation coefficient) [24] and the pacing design (i.e., asymmetry characteristics of the race) [3] were compared. The equation of Kendall’s *τ*: (3)$$K\left(vi,vj\right)=K\left(vj,vi\right)=\left\{\begin{array}{c} 1\;if\;\;i<j\;and\;vi<vj\\ 0\;if\;vi=vj\\ -1\;if\;\;i<j\;and\;vi>vj \end{array}\right.\;$$
(4)$$\tau =\frac{2}{n\left(n-1\right)}\;\underset{i<j}{\sum }K\left(vi,vj\right)$$

*vi* = ith value of a speed; *vj* = *j*th value of a speed; *i* < *j* = *i* indicates a period of time prior to *j*; sum being performed over the *n*(*n* − 1)/2 distinct unordered couples of indices {*i*, *j*}, so that τ takes values in between −1 and 1.

Furthermore, we used the Pearson median skewness to calculate the skewness value of the speed distribution as 3 (mean − median)/SD. Skewness is a measure of the asymmetry of the probability distribution of a real-valued random variable about its mean. A negative coefficient indicates a distribution shifted to the right of the median and thus a distribution tail spread to the left.

Its value can be positive, negative, or undefined. A positive skewness means that the mean is greater than the median, while a negative skewness means the mean is less than the median. In this case, it means that the marathon runner covered more kilometers above the final average speed due to the decrease in his speed in the last part of the race [3].

The variability of the signal was determined by calculating the coefficient of variation using this formula [25]:(5)$$\mathrm{Coefficient}\;\mathrm{of}\;\mathrm{variation}=\frac{\mathrm{std}}{\mathrm{mean}}\times 100$$

Where std = the standard deviation of the signal over the marathon; mean = the average of the signal over the marathon.

## 3. Results

All subjects completed the race, and four of them ran their personal best times despite the weight of the devices (Table 1). The final blood lactate value was equal to 2.6 ± 0.5 mM vs. 1.4 ± 0.5 mM after the warm-up (*p* < 0.05).

### 3.1. Statistical Characteristics of the Variables throughout the Marathon (Speed, Heart Rate, and Stride Length)

The average time to complete the race was 196 ± 20.3 min. Table 3 shows the descriptive data of speed, heart rate, and stride length for the marathon. The average speed was 13.0 ± 1.4 km.h^−1^, the heart rate average was 155 ± 8 bpm, and the stride length average was 1.3 ± 0.17 m. The coefficient of variation speed in this study was averaged at 6.0 ± 1.4%.

All subjects ran a large positive gap race, as indicated by a negative Kendall’s τ for speed and stride length (SL) in contrast to heart rate (HR), which had a positive Kendall’s τ (Table 4). Speed and SL decreased significantly throughout the race, given this overall trend. In addition, speed, HR, and SL exhibited negative skewness (Table 4).

### 3.2. Entropy on the Marathon

The variable that provides the highest entropy during the marathon is the stride length (Table 5). However, neither the stride length entropy, nor the heart rate and speed one were correlated with the performance following the significance set at *p* < 0.05. Indeed, the H entropy of stride amplitude was almost correlated with performance (r = −0.796, *p* = 0.060).

The distance at which the *H*_SL%_*H*_max_ falls was highly variable (30.3 ± 8.0 km), at 2 h 21 min ± 43 min. The distance at which the *H*_SL%_*H*_max_ decreases is highly correlated with performance (r = −0.981, *p* = 0.001) (Figure 2). During the marathon and until the runner hits the wall, the runner’s entropy fluctuates around 50 ± 6% of the maximum entropy.

The distance at which the *H*_SL%_*H*_max_ decreases is highly correlated with the distance at which RPE exceeds 15 (running becomes “very hard”) (r = 0.903, *p* = 0.014) (Figure 3). The distance at which RPE exceeds 15 is highly correlated with the distance at which speed falls (r = 0.910, *p* = 0.012) (Figure 4). However, there is no significant correlation between the distance at which the speed drops and the one at which *H*_SL%_*H*_max_ decreases (r = 0.784, *p* = 0.1).

## 4. Discussion

In this study, our primary objective is to analyze the relationship between marathon time and the kilometer at which a decline in stride length (SL) reserve entropy occurs. This decline is identified when the entropy falls below the mean value minus two standard deviations. We observe that elite athletes experience this decline over longer distances. A decrease in entropy indicates a flatter probability distribution of different states, transitioning from a dominant state in SL (more probable) to a state characterized by increased variability.

Consequently, the greater the loss of complexity during the marathon, the better the performance. Healthy physiological systems exhibit complex and predictable fluctuations, while systems under greater relative stress tend to demonstrate reduced complexity [26]. 

Additionally, we examine the relationship between the kilometer at which RPE exceeds a value of 15 (hard), the kilometer at which SL normalized entropy decreases, and the kilometer at which speed declines. In both cases, we observe a significant positive correlation.

The marathon race remains unpredictable, unlike the half marathon, where the final performance is more predictable due to the absence of the marathon wall [27]. Although not all runners experience hitting the wall, it is a risk encountered by most marathon runners at some point in their race [28]. In this pilot study, our aim is to investigate whether hitting the wall corresponds to the moment when physiological and mechanical parameters fall below a threshold associated with maximal entropy, as measured during a maximal test. We specifically focus on examining SL, heart rate (HR), and speed, which are now readily available thanks to the use of cardio-GPS and accelerometers worn by many runners.

The main purpose of this study was to examine the hypothesis that the marathon wall corresponds to the loss of stride length entropy and an increase in RPE. The second aim was to examine a possible relationship between performance (marathon time) and the distance at which entropy drops. The results showed that stride length had the highest entropy among the variables, and a reduction in its entropy to less than 50% of its maximum value (H = 3.3) was strongly associated with the distance (between 22 and 40) at which participants reported “hard exertion” (as indicated by an RPE of 15).

Interestingly, our results indicated that heart rate (HR) could not be utilized as an effective indicator of the marathon wall, as it displayed a continuous increase throughout the race, consistent with previous findings [4]. Furthermore, our findings confirmed that the least efficient runners exhibited the least asymmetric distribution of speed and HR [3,4]. HR demonstrated a consistent upward trend, while speed declined, and stride length remained relatively stable until the 26 km mark [29,30]. The coefficient of variation for speed in our study averaged 6.0%, which was lower compared to previous research that reported values ranging from 16.9 ± 6.4% [25].

Even when marathon runners’ self-pace their race based on the rating of perceived exertion (RPE), instead of heart rate (HR) as instructed in this experiment, a significant decrease in speed is commonly observed after the 26th kilometer on average [4]. This decline has been attributed to a lack of speed variability and the inability to recover without compromising average speed [21]. This suggests that runners may lack the ability to effectively pace their running, possibly due to insufficient information from physiological and mechanical cues. Although our marathoners were proficient recreational runners with a time of 3 h and 30 min, it is plausible, as suggested by previous authors [8,11], that experienced marathoners possess the ability to predict their pace, as proposed by the “teleoanticipation theory”. This theory posits that pace is pre-determined in advance and continually adjusted and processed by the brain, incorporating feedback from various central and peripheral signals to prevent catastrophic failure of physiological systems [8,11]. Furthermore, it has been demonstrated that the brain employs a pacing algorithm for a specific event with a defined endpoint, incorporating prior knowledge of distance, duration, and pacing strategies to optimize performance [31]. Additionally, subconscious homeostatic control systems that modulate power output based on feedback from physiological systems may also be interconnected with conscious emotional responses that generate subjective “feelings” [32]. The field of information theory, first introduced by Shannon in 1948 [13], was developed to address the challenge of transmitting information efficiently and reliably through communication channels, balancing high transmission rates with minimal errors. The primary objective of this theory was to devise methods for transmitting information through noisy channels with optimal efficiency and reliability. In our study, we discovered that stride length, rather than physiological signals such as heart rate, exhibited significant variations in information content (reducing to less than 50% of its maximum value) that were strongly associated with the onset of fatigue. This was evidenced by a decline in speed and an increase in the Rating of Perceived Exertion (RPE) between the half marathon and the 40th kilometer, relative to the individual’s performance level.

During an exhaustive race, the body is subjected to significant stress, which can lead to significant fluctuations in heart rate and stride frequency. These fluctuations can lead to an increase in the entropy of these variables. To quantify the entropy of heart rate and stride frequency, signal processing and information theory techniques can be used. In information theory, Claude Shannon’s concept of entropy is a measure of the randomness or uncertainty of a system [13]. It indicates the amount of information required to describe the system, with higher entropy indicating greater unpredictability. 

Elevated entropy in heart rate and stride frequency may suggest that the body is exerting more effort to sustain performance while maintaining variability. It may also indicate increased uncertainty and variability in the physiological response to the race. Conversely, lower entropy values may signify more stable and predictable patterns of response. Overall, the concept of entropy provides a valuable framework for comprehending the complexity of physiological systems during an exhaustive race and may provide insights into the factors influencing performance and recovery.

The concept of “information” shares similarities with probability; it is a general, qualitative, and subjective concept. For instance, the psychophysiological rating of perceived exhaustion (RPE) scale developed by Borg, or the feeling of fatigue described by Gibson et al. [32], stems from conscious awareness of changes in subconscious homeostatic control systems. This awareness arises due to temporal disparities between the subconscious representations of these homeostatic control systems in neural networks induced by activity level changes.

These mismatches are perceived by brain structures involved in consciousness as a sensation of fatigue. In our study, we observed that the occurrence of the “marathon wall” coincides with a reduction in physiological information, specifically a decrease in entropy. If this holds true for heart rate data as well, future iterations of cardio-GPS devices could incorporate this entropy drop as an indicator of suboptimal pacing long before a decrease in speed occurs. This approach would enable the identification of which information in the signals corresponds to specific performance outcomes. For instance, we could calculate the Shannon entropy of heart rate or stride frequency time series data, allowing us to measure the unpredictability or randomness of these variables over time across a broad range of runners of varying ages, genders, and performance levels (relative to the world record in their respective categories).

Our findings revealed that stride amplitude exhibits the highest entropy during the marathon. The fact that heart rate demonstrates low entropy but a general upward trend during the race indicates that heart rate is not a rapid adjustment variable during the marathon, as previously reported [3,4,33]. Importantly, our results demonstrate a strong correlation between the distance at which stride length entropy decreases below 50% of its maximal value (referred to as the entropy maximal reserve, EMR) and marathon performance, as well as the distance at which RPE exceeds a value of 15 (indicating a very difficult race).

Therefore, greater loss of complexity during the marathon is associated with improved performance, and the 50% EMR threshold could serve as an indicator of hitting the wall. Healthy physiological systems exhibit predictable and complex fluctuations, while systems under higher relative stress tend to display reduced complexity [26].

This “loss of complexity” phenomenon with aging has been observed across various outcomes, including electroencephalogram, gait, and muscle torque. Significantly, it is interpreted as a decline in adaptability and a narrowing of the system’s responsiveness [34].

Running speed is determined by the combination of cadence and stride length. Humans adjust their speed primarily through changes in stride length rather than cadence [35]. While cadence is influenced by leg length for a given speed, stride length also relies on muscular strength. During running, unlike walking, there is a flight phase where both feet are off the ground. This enables runners, particularly experts, to overcome gravity with their kinetic energy surpassing their potential energy. The ratio of these energies gives rise to the well-known Froude number, which distinguishes walking from running. Hence, it is possible for individuals with smaller stature, such as the Ethiopian champion Haile Gebrselassie, to possess a significant stride length. This is contingent upon their muscle strength [36,37].

Murray et al. (2017) demonstrated that running complexity is influenced by speed and lactate accumulation and can serve as an explanatory variable for lactate threshold and maximal aerobic power [38]. Although blood lactate concentration was not measured in our current study, the pace maintained during the marathon remained below the ventilatory threshold estimated using the RABIT^®^ test. This test has been validated for estimating VT2 from moderate-paced running [18].

## 5. Conclusions and Study Limitations

With the availability of various tools to measure running parameters, such as stopwatches, heart rate monitors, and speed trackers using GPS or accelerometers, runners now have the means to monitor their cadence and stride amplitude. The objective of this study was to explore the potential of detecting muscle fatigue by examining the decrease in stride amplitude variability, which serves as the speed adjustment variable in human running, including marathon running on flat terrain. It is important to note that this is a pilot study conducted with a small sample size of only six male subjects. Therefore, caution should be exercised in interpreting the results, as they may not be generalizable to larger populations. Nevertheless, the promising findings related to analyzing pacing strategy and the onset of fatigue through a reduction in stride length entropy below 50% of maximum entropy motivate further application of this method to a larger group of marathon runners, especially considering the increasing number of individuals utilizing reliable cardio-GPS monitors. However, it is worth mentioning that measuring stride length necessitates the use of a belt equipped with an accelerometer, which offers higher accuracy compared to measuring cadence at the wrist. Additionally, it is important to note that entropy analysis can only be performed post-marathon and not during the race. The significance of the current study lies in providing runners with a novel perspective on understanding “performance” by recognizing pacing intelligence as an additional component alongside time. Many marathon runners often question whether they could have performed “better” and how to achieve it. The marathon continues to captivate and intrigue due to its long and intense nature.

## Figures and Tables

**Figure 1 entropy-25-01119-f001:**
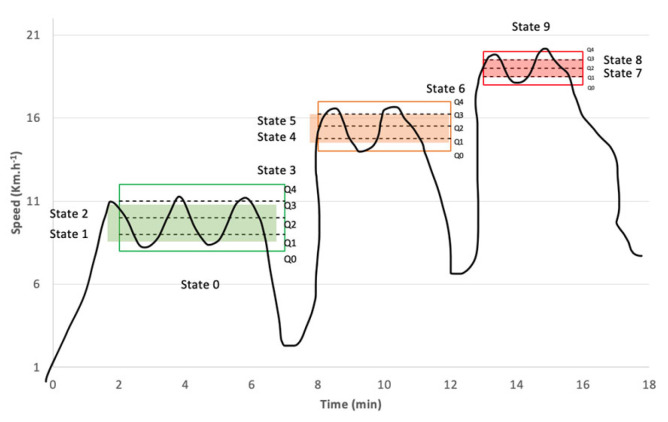
Distribution of a subject’s speed according to the 9 possible states (N = 9). The green box represents the easy sensation, the orange box the medium sensation, and the red box the hard sensation.

**Figure 2 entropy-25-01119-f002:**
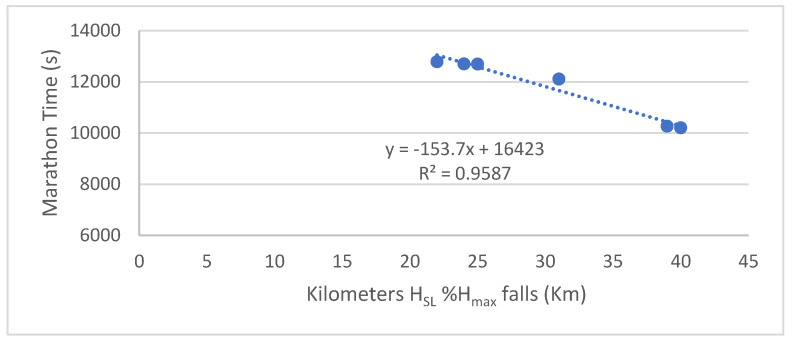
Relationship between marathon time (s) and distance at which the *H*_SL_%*H*_max_ falls (km). *H*_SL%_*H*_max_ is the stride length entropy in % of *H*_max_.

**Figure 3 entropy-25-01119-f003:**
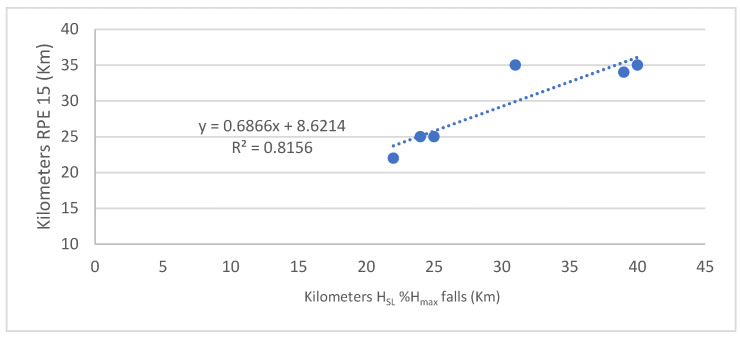
Relationship between the distance (km) at which RPE exceeds 15 and the distance (km) at which the *H*_SL_%*H*_max_ falls. *H*_SL_%*H*_max_ is the stride length entropy in % of *H*_max_.

**Figure 4 entropy-25-01119-f004:**
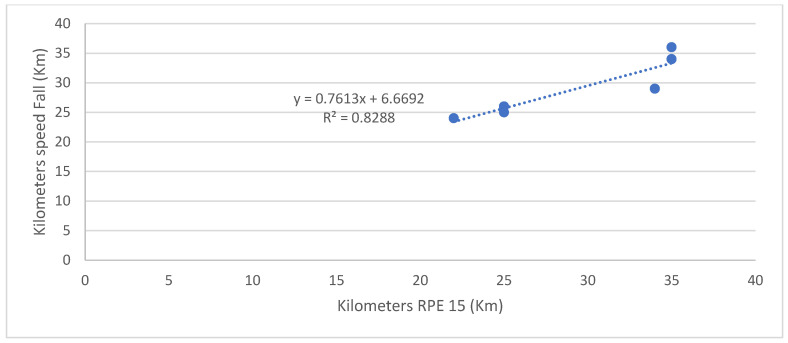
Relationship between the distance (km) at which RPE exceeds 15 and the distance (km) at which the speed falls.

**Table 1 entropy-25-01119-t001:** The age of the subjects, their personal record, and the year of this performance. * Subjects who beat their personal best during the Lake Annecy Marathon.

Runners id	Age (Years)	Fastest Marathon Time (Years)	Lake Annecy Marathon (2019)
1	34	02 h 55′03″ (2018)	02 h 50′00″ *
2	33	02 h 53′43″ (2017)	02 h 51′36″ *
3	23	03 h 10′12″ (2019)	03 h 21′40″
4	42	03 h 32′23″ (2018)	03 h 31′27″ *
5	44	03 h 35′59″ (2017)	03 h 31′34″ *
6	47	03 h 12′46″ (2018)	03 h 32′58″

**Table 2 entropy-25-01119-t002:** The 9 states of each variable are calculated using the RABIT^®^ test, with a summary of the RPE’s and associated HR: heart rate (bpm), speed (km.h^−1^), and stride length (m) (example for the runner n°3).

States	Speed	HR	Stride Length
1	x < 12.4	x < 145	x < 1.10
2	12.4 ≥ x < 12.9	145 ≥ x < 150	1.10 ≥ x < 1.15
3	12.9 ≥ x < 13.6	150 ≥ x < 154	1.15 ≥ x < 1.21
4	13.6 ≥ x < 14.2	154 ≥ x < 161	1.21 ≥ x < 1.36
5	14.2 ≥ x < 14.8	161 ≥ x < 164	1.36 ≥ x < 1.41
6	14.8 ≥ x < 16.5	164 ≥ x < 167	1.41 ≥ x < 1.52
7	16.5 ≥ x < 16.8	167 ≥ x < 170	1.52 ≥ x < 1.62
8	16.8 ≥ x < 17.3	170 ≥ x < 176	1.62 ≥ x < 1.71
9	x ≥ 17.3	x ≥ 176	x ≥ 1.71

**Table 3 entropy-25-01119-t003:** Performance during the marathon race. Speed: speed for the marathon (km.h^−1^), HR: heart rate (bpm), SL: stride length (m).

Runners id	Marathon Time		Speed	HR	SL
1	2 h 50 min 00 s	Mean	14.9	146	1.5
		SD	0.6	9	0.2
		Coefficient of variation	4.3%	6.4%	13%
2	2 h 51 min 36 s	Mean	14.8	150	1.5
		SD	1.8	6	0.1
		Coefficient of variation	5.1%	3.7%	12%
3	3 h 21 min 40 s	Mean	12.5	159	1.1
		SD	0.8	5	0.1
		Coefficient of variation	6.9%	3.5%	7.7%
4	3 h 31 min 27 s	Mean	12.1	149	1.4
		SD	1.3	7	0.2
		Coefficient of variation	6.7%	4.6%	16%
5	3 h 31 min 34 s	Mean	12.0	167	1.2
		SD	0.7	6	0.1
		Coefficient of variation	5.8%	2.8%	11.3%
6	3 h 32 min 58 s	Mean	11.9	159	1.2
		SD	0.9	5	0.1
		Coefficient of variation	7.5%	3.3%	7.7%

**Table 4 entropy-25-01119-t004:** Trend and asymmetry (skewness: SK) characteristics of speed, HR (heart rate), and stride length in the marathon race.

Runners id		Speed	HR	Stride Length
1	Kendall Tau	−0.456	0.506	−0.464
	*p*-value	0.001	0.001	0.001
	SK Pearson	−0.555	−0.193	−0.584
2	Kendall Tau	−0.577	0.417	−0.400
	*p*-value	0.001	0.001	0.001
	SK Pearson	−0.534	−0.956	−0.101
3	Kendall Tau	−0.462	0.381	−0.357
	*p*-value	0.001	0.001	0.001
	SK Pearson	−0.160	−1.443	0.278
4	Kendall Tau	−0.306	0.374	−0.390
	*p*-value	0.001	0.001	0.001
	SK Pearson	−0.551	−0.983	−0.797
5	Kendall Tau	−0.610	0.385	−0.466
	*p*-value	0.001	0.001	0.001
	SK Pearson	−0.190	−0.644	−0.27
6	Kendall Tau	−0.632	0.126	−0.513
	*p*-value	0.001	0.001	0.001
	SK Pearson	−0.695	−1.566	−1.161

**Table 5 entropy-25-01119-t005:** Mean and standard deviation of the entropy of each kilometer of the marathon speed, HR (heart rate), and SL (stride length).

Runners id		Entropy Speed	Entropy HR	Entropy Stride Length
1	Mean	1.22	1.18	1.44
	Standard deviation	0.47	0.51	0.48
2	Mean	0.77	0.57	1.60
	Standard deviation	0.60	0.38	0.30
3	Mean	0.81	0.65	1.33
	Standard deviation	0.65	0.56	0.60
4	Mean	0.81	0.50	1.32
	Standard deviation	0.58	0.47	0.54
5	Mean	0.23	0.23	0.96
	Standard deviation	0.43	0.43	0.75
6	Mean	1.06	0.23	1.16
	Standard deviation	0.57	0.42	0.57

## Data Availability

The data presented in this study are available on request from the corresponding author.

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
