# Peer review of "A Pilot Study Using Entropy for Optimizing Self-Pacing during a Marathon"

_entropy, 2023, doi:10.3390/e25081119_

Round 1
Reviewer 1 Report
In this manuscript the authors explore the relationship between self-pacing, perceived exhaustion, and performance in marathon runners. The study aims to understand how runners can effectively pace themselves based on physiological and mechanical signals.
This pilot study analyses 3 variables: heart rate (HR), speed (V) and stride length (SL), recorded in non-elite runners during a marathon. These variables are measured using 5 s data averages.
In addition to these 3 variables, the authors also record the VO2 oxygen uptake and the rate of perceived exertion (RPE). It is not clear from the study whether this RPE is a subjective scale perceived by the runner or whether it is calculated/estimated based on the different variables. This should be better clarified in the manuscript.
For each kilometer of the marathon, the normalized Shannon Entropy of each of these 3 variables is calculated. However, according to those explained in section 2.5, it is very difficult to know how this entropy is calculated. As I understand, in the text it is an example of how it is calculated for speed, and it seems that the values are quantified in 9 states depending on the RPE of the subject (I understand that they are the RPE values obtained in the RABIT test*). Then the entropy of this data is calculated in 1 km sections. Perhaps to compare the same amount of data across all subjects, it would be preferable to calculate the entropy in 5-minute stretches rather than 1 km stretches.
* If these 9 states of each variable are calculated with the RABIT test, an additional results subsection should be added with a summary of the RPE’s and associated HR, V and SL values of each subject.
Additionally, the coefficient of variation, the Kendall’s tau rank correlation and asymmetry is calculated to evaluate the trend of the HR, V, and SL during the race. Here again it is not clear whether these coefficients of variation are calculated using all values of RH, V and SL, i.e., the values calculated every 5 seconds, or the average values calculated every km. Please clarify this aspect.
The authors report firstly a coefficient of variation between 4.3 and 7.5 % for V, between 2.8 and 6.4% for RH and between 7.7 and 13% for SL. As for the trend, a positive gap is reported for V and SL (these variables decrease throughout the race) and a negative gap for HR (this variable decreases throughout the race).
No significant correlations are observed between mean entropy (or sum of entropies per km) and marathon performance, although for SL the correlation is almost significant, indicating higher entropy (greater variation of SL states?) in athletes with better performance.
In addition, the authors analyze several relationships.
Firstly, they analyze the relationship between marathon time and the km at which there is a fall in the normalized entropy of the SL (fall: according with section 2.4 this drop is determined when the entropy decreases below the mean value minus 2 standard deviations), and it is observed that in athletes with better performance this decrease occurs at longer distances. What does this mean? If the entropy decreases, it means that the probability distribution of the different states is flatter, that is, it goes from an SL in which some states predominate (they are more probable) to another in which there is more variability. What physiological significance can this have? Add discussion about this result.
Secondly, they analyze the relationship between the km in which the RPE exceeds the value of 15 with (1) the km at which there is a fall in the normalized entropy of the SL and (2) the km at which there is a fall in the speed. In both cases it is observed a positive significant correlation.
The authors should try to point out and make explicit the conclusions of the study. The discussion and conclusions sections should be completely rewritten, focusing on discussing the results shown in the study. The conclusion in abstract only states: "These findings suggest that integrating stride length's Entropy feedback into new cardio GPS watches could improve marathon runners' performance", but the extent and specific mechanisms of this potential improvement are not explicitly mentioned. The fall in the SL entropy parameter seems to be linked to the arrival of the "marathon wall", but how can this entropy parameter that must be integrated in the watches improve the performance of the runners? What entropy values are indicated to improve the runner's performance?
Finally, the authors should point out in the results/conclusions of this study should be interpreted with caution for the small sample size and may not be generalizable to larger populations.
Minor:
Page 2, lines 47-49: “Recent studies have analyzed the continuous physiological response and RPE during a marathon and have shown that the ratio between RPE and speed, heart rate, and VO2 decreases similarly for all recreational runners [4].”
Define the “Rabit” test: RABIT® (Running Advisor Billat Training). Rabit should be capitalized as it is an acronym.
Equations (1)/(2): Change log by log2 in equation (1), or log2 by log in equation(2)
Equation (3): The operator K() has not been defined
Table 4: In this table it is shown the sum of the entropy of each km of the marathon. If the entropy is normalized, it should have values between 0 and 1, however in this table the maximum values exceed the km of the marathon. Please clarify the data reported in this table. It would also be preferable to show the average entropy rather than the sum of the entropies of all km.
Figure2/Text: The R^2 value showed in the figure is not equal to (r)^2, being r the value showed in the text.
The “Supplementary Materials/Author Contributions/Funding/Institutional Review Board Statement/Data Availability Statement/Acknowledgements/Conflicts of Interest” sections should be completed with data from the study.
Section 6 (page 12) is no necessary. It should be deleted.
Reviewer 2 Report
The submitted manuscript describes a pilot study on the optimization of self-pacing during a marathon. The results show that the stride length had the highest entropy among the variables and a reduction in its entropy to less than 50% of its maximum value (H=3.3) was strongly associated with the distance
(between 22 to 40) at which participants reported a "hard exertion" (as indicated by an RPE of 15). Although the presented results are valuable, there are a few points the authors should address.
1. Abstract, lines 13-14: I would paraphrase "A new category of marathon runners with limited experience hits “the wall” which refers to a significant decrease in speed and an increase in perceived exhaustion (RPE)." as e.g. "A new group of marathon participants with minimal prior experience encounters the phenomenon known as "hitting the wall," characterized by a notable decline in velocity accompanied by the heightened perception of fatigue (Rate of Perceived Exertion, RPE)."
2. Introduction: lines 50-79 need proofreading, especially introducing the term information entropy (lines 69-79).
3. Materials and methods:
a) including female subjects could provide information about the influence of gender on the "hitting the wall" phenomenon and appropriate entropy indicators.
b) Did you consider determining which muscles are activated during the marathon?
c) Did you consider the measurements of SmO2 (muscle oxygen saturation) in all muscles that are activated while running a marathon?
d) Formatting of subsubsections - you can use numbering for subsubsections and proper formatting according to the template.
e) Lines 175-177: minor formatting issues (subscripts)
f) The number of recruited subjects (only 6 male) is quite small
4. Discussion:
a) The text needs proofreading.
b) There is no mention of limitations of the study, such as the limited number of subjects (only 6 male), using only four physiological signals (VO2max, HR, RFE, SL)
c) Sentence "Numerous publications demonstrated the effectiveness of multimodality for machine learning." (lines 377-378) is vague without citing appropriate references.
d) Presented results should be critically analyzed and compared to similar studies.
5. Conclusion: The conclusion lacks any reference to the limitations of the study.
6. Statements of authors:
a) Supplementary Materials: Do you intend to publish the dataset or attach any additional material to the manuscript?
b) Declare the contribution of each author according to CRediT taxonomy
c) Is your study funded by an agency or institution? If so, indicate an appropriate grant number and name of the funder.
d) Institutional Review Board Statement: mention the approval by the
institutional review board (CPP Sud-Est V, Grenoble, France; reference: 2018-A01496-49)
e) Data availability statement: see a) or give the reason for the (un)availability.
f) Acknowledgments: if there are any people or entities that helped in your study.
g) Conflict of Interest: declare conflicts or use the statement "The authors declare no conflict of interest".
The quality of English is sufficient to understand the study background, methods, material, results, and conclusion, but it could be improved by proofreading the manuscript and a consistent style.
Round 2
Reviewer 2 Report
The authors have addressed my comments, except for the comment concerning the institutional review board - this has not been addressed yet.
Lines 285-291: Emphasis not needed.
The text could be slightly proofread for increasing clarity and integrity.
Author Response
First, we'd like to thank the experts for taking the time to help us improve this manuscript.
We've added the "Institutional Review board statement" section and improved the English for the introduction, discussion and conclusion sections.